# Brain Organization and Human Diseases

**DOI:** 10.3390/cells11101642

**Published:** 2022-05-14

**Authors:** Tamar Sapir, Dalit Sela-Donenfeld, Maayan Karlinski, Orly Reiner

**Affiliations:** 1Departments of Molecular Genetics and Molecular Neuroscience, Weizmann Institute of Science, Rehovot 7610001, Israel; maayan.karlinski@weizmann.ac.il; 2The Robert H. Smith Faculty of Agricultural, Food and Environmental Sciences, School of Veterinary Medicine, The Hebrew University of Jerusalem, Rehovot 7612001, Israel; dalit.seladon@mail.huji.ac.il

**Keywords:** cortical development, neuronal migration, holoprosencephaly, lissencephaly, forebrain, telencephalon, signaling factors

## Abstract

The cortex is a highly organized structure that develops from the caudal regions of the segmented neural tube. Its spatial organization sets the stage for future functional arealization. Here, we suggest using a developmental perspective to describe and understand the etiology of common cortical malformations and their manifestation in the human brain.

## 1. Introduction

Understanding the origins of higher cortical functions and their malfunction in developmental diseases requires delineating how brain areas are formed during development. It has been long appreciated that the human brain is composed of defined areal and laminar structures that differ in composition [1,2,3,4]. These areas are specified during embryonic development due to orchestrated evolutionary conserved processes that sculpt the brain. The limited access to human embryos, ethical considerations, and the lack of suitable in vitro human model systems make human brain organization a complex topic to study. Today’s information is an accumulation of data from different model organisms and, in part, from human embryos. This review explores major developmental events leading to the formation of regional forebrain domains and cortical layers. It examines how these processes contribute to our understanding of the pathophysiology of common human brain developmental disorders.

### 1.1. Forebrain Induction and Patterning

The first indication of the developing human central nervous system (CNS) appears in the third week of human embryonic development in a process termed neurulation. A plate of thickened ectoderm appears in the mid-dorsal region of the tri-laminar embryo, the neural plate. The formation of the anterior neural plate, which will give rise to the forebrain, is induced by signals from the prechordal plate (PrCP), a cell population that originates from the mesendoderm and migrates rostrally from the primitive node along the midline between the ectoderm and the endoderm layers. In a series of extensive morphogenetic movements, the neural plate bends inwards to generate a tubular structure with open neural folds, then brought together at the dorsal midline to generate the neural tube (scheme in Figure 1). While neural plate induction occurs along the entire rostral–caudal axis at the dorsal midline, the PrCP is responsible for the rostral neural plate cells to adopt an anterior fate and antagonizes the activity of caudalizing factors that are secreted at more posterior mesodermal domains [5].

By the end of the fourth week of development, the rostrally positioned primary vesicles become noticeable and form the morphological segmentation that will become the distinct functional units of the brain [8]. The primary vesicles include three vesicles: prosencephalon, mesencephalon, and rhombencephalon (which will later be divided into the metencephalon and myelencephalon(. A week later, the primary prosencephalus becomes subdivided into three secondary vesicles: at the anterior end of the neural tube, the left and right telencephalic vesicles, and a more caudal diencephalon vesicle (Figure 1b). The telencephalon will eventually give rise to the left and right cerebral hemispheres, the olfactory bulbs, and the hippocampus.

The diencephalon will generate the optic vesicles that become the retina, thalamus, and hypothalamus, which develop from dorsal or ventral domains. Notably, the current view supports the hypothesis that the segmented neuroaxis evolved from a series of many more repeated segments than those morphologically evident, termed metameric units. This organization paradigm has early evolutionary origins, is already apparent in the embryonic insect, and is dissected by homologous segment identity genes in vertebrates. The prosomeric model depicts a segmental subdivision of the body axis, based on gene expression patterns, to the so-called “prosomere,” separated by nonidentical boundaries [9]. Metamerism was suggested to extend to the nervous system’s anterior parts, including the caudal forebrain, but is missing in the prosencephalon (telencephalon and hypothalamus) [9].

### 1.2. Signaling Factors That Control Forebrain Development

The organization along the anterior–posterior (AP) and dorsal–ventral (DV) axes of both the spinal cord and the anterior structures of the CNS is governed by the production of instructive molecules—morphogens—that are expressed by small cell populations, often referred to as patterning centers or organizers. A fundamental question is how positional information is conveyed and interpreted by the cells to express subsets of transcription factors that will execute the developmental fate. A model for the interpretation of morphogens areal information into distinct cell fates was proposed by Wolpert (the French flag model) [10]. According to this model, effective patterning driven by morphogen gradients will obey three required elements: (i) polarity, (ii) differential response of cells, and (iii) at least one spontaneous self-limiting reaction. The robustness of morphogen patterning requires a buffer mechanism that ensures a reproducible response [11].

The anterior neural ridge (ANR) is an example of an organizing center that governs forebrain patterning via positional information. The ANR is a temporary structure located at the most-rostral edge of the neural plate (Figure 1a). The ANR is initially necessary for prosencephalon induction. Moreover, it also sets a “protomap” upon which the prosencephalon is programmed to subdivide into the telencephalon, optic vesicles, and diencephalon domains, based on the regionalized expression of regulatory genes [12,13,14,15,16]. The ANR, which later becomes the anterior telencephalon, is the source of soluble morphogens that belong to the fibroblast growth factor (FGF) family, such as FGF8 and FGF17 [17,18,19]. The instructive role of FGF8 signaling in shaping the anterior–posterior neocortical map and its particular role in the shaping of the most anterior part of the telencephalon were highlighted by a detailed analysis of the arealization of the forebrain in mutant mice embryos with reduced FGF8 levels (Fgf8 neo/neo and Fgf8 neo/null hypomorphic mutants [20,21,22,23]).

Similarly, ablation of the ANR or loss of Fgf8 in the anterior neural plate cells of zebrafish embryos resulted in a distorted telencephalon with abnormal morphology, altered gene expression, and axonal misprojection [24]. Furthermore, in utero electroporation of a soluble FGF8 receptor that competed with the endogenous FGF8 receptors resulted in shifts in telencephalic boundaries [14]. Conversely, forced expression of Fgf8 in E11.5–E12.5 mouse embryos in the anterior cortical primordium resulted in enlarged cortical areas with anterior identity and a reciprocal reduction of posterior fates [25]. Notably, Fgf17 function was found to be similar to that of Fgf8, with a more selective role in regulating the properties of the dorsal frontal cortex [26].

Following the formation of the left and right telencephalic vesicles, the mediolateral axis of their dorsal cortical primordium develops into two signaling centers: the hem and the anti-hem. Both centers flank the cortical neuroepithelium and express different types of signaling molecules. The cortical hem expresses Wingless (Wnt) genes (Wnt2b, 3a, 5a, 7b, 8b) and bone morphogenetic protein (BMP) genes (Bmp2, 4, 5, 6, 7) and serves as the organizer of the hippocampus. Indeed, the induction of ectopic hem structures adjacent to Lhx2 null cells can produce multiple hippocampal fields [27,28]. The anti-hem is located at the pallial/subpallial boundary and expresses secreted signaling molecules, including FGF7 and three epidermal growth factor (EGF) family molecules, yet its organizing functions are not yet fully understood [28].

In addition to FGF, BMP, and Wnt signals governing the AP and mediolateral patterning of the dorsal telencephalon, additional signaling factors participate in the patterning of the ventral and the lateral telencephalon. Sonic Hedgehog (Shh), a morphogen known for its graded function in the ventral spinal cord, has been shown to work in two phases during forebrain development. In the early stage, Shh is expressed in the PrCP and mediates the initiation of forebrain development from the anterior neural plate as well as the subdivision of the anterior prosencephalon into left and right vesicles. Next, the early Shh signal triggers a secondary induction activity in the forebrain, where it will sequentially specify the progenitors of the medial ganglionic eminence (MGE) and, later, of the lateral ganglionic eminence (LGE) (Figure 1c) [29].

Retinoic acid (RA), a metabolite of retinol (vitamin A), functions as a ligand for nuclear retinoic acid receptors (RARs), which regulate the expression of numerous downstream target genes during both early embryogenesis and adulthood. Early in development, RA appears in a morphogenic gradient (high caudally, low rostrally) established by an interplay between diffusion gradients and localized RA metabolism, thereby allowing RA to control the patterning of various structures [30]. RA is locally synthesized in the telencephalon neuroepithelium in the early stages of forebrain development. Prevention of RA signal transduction will significantly impair forebrain development by reducing neurogenesis and cell proliferation and increasing cell death [31,32]. Later, RA is required to properly differentiate GABAergic and dopaminergic neurons in the striatum and neocortical lamination and for radial migration during mouse corticogenesis [33,34].

An essential principle in forebrain induction and patterning is that signaling molecules, such as those mentioned, do not work independently but rather display an intricate regulatory network. Different secreted morphogenetic molecules in and around the patterning centers govern direct or indirect interactions between multiple signaling pathways and their downstream transcription factors. For instance, Fgfs, expressed at the ANR, are regulators of mid-line patterning and antagonize Wnt and BMP of the cortical hem [35]. At later stages, the proliferative activity of RA involves a crosstalk of Shh and FGF signaling [36].

Finally, while the local AP planar induction sets up a transverse regionalization of the major forebrain parts, DV patterning along the neural tube is also fundamental for defining longitudinal zones. The best-studied DV patterning of cellular and molecular events in the CNS is that that shapes the hindbrain and the spinal cord DV axis, in which distinct neuronal subtypes emerge from pre-patterned progenitor domains at a defined location along the DV axis [37,38,39,40] to govern sensory and motor neural activities, respectively. Although less knowledge exists on the anterior neural tube, recent data indicate that the dorsal or ventral fates of neural progenitors are similarly patterned in the human forebrain [41,42].

### 1.3. Neuronal Migration and Cortical Layers

The cerebral cortex has an intrinsic functional architecture critical for its proper performance. It is derived from the dorsal and ventral telencephalon. The dorsal telencephalic domain gives rise to the significant neuronal population that will occupy the brain, the excitatory neurons, while the ventral telencephalon includes the ganglionic eminences (GE) from which the minor population of GABAergic interneurons originates. The emergence of different functional areas results from orchestrated events discussed elsewhere [43,44].

Cortical development involves several modes of migration during which cells move away from their place of birth. Migratory events are generally categorized based on their general movement pattern (radial or tangential) or by the characteristic morphological or dynamic behavior of the migrating cells (multipolar migration, somal translocation). An important example of how cell migration shapes the brain is that of the early-born neurons, the Cajal–Retzius cells (CRs). CRs are one of several transient cell populations in the developing brain [45] and spread along the superficial preplate and marginal zone throughout corticogenesis. They are well known for their role as regulators of radial migration by the secretion of an instructive extracellular matrix protein, Reelin, and their direct interaction with the end-feet of the radially migrating cells [46,47]. Additionally, CRs from different origins spread in a complementary manner throughout the cortical surface and set boundaries of functional areas within the cortex [45].

The primordial progenitors of the cortex are the neuroepithelial cells (NECs). NECs are elongated cells that span the entire thickness of the neuroepithelium from the ventricular (apical) surface to the laminal (basal) side [48,49] (Figure 2). During proliferation, the nuclei move within the cells’ cytoplasm, traveling large distances in synchronization with the cell cycle phase [50,51]. Nuclei undergoing mitosis occupy the apical surface. This is followed by an apical-to-basal motion of the daughters’ nuclei and progression of the G1 phase. The S phase occurs as the nuclei reach the basal side of the cells, and G2 occurs during basal-to-apical motion.

This cyclic motion, termed interkinetic nuclear migration or motion, was first discovered by Schaper in 1897 and re-discovered by Sauer in 1935 [52,53,54]. Later, the neuroepithelium gives rise to and is eventually replaced by apical radial glia (aRG) that divide symmetrically to increase the pool of progenitors. Alternatively, asymmetrical divisions result in new types of progenitors, including intermediate progenitors (IPs), basal radial glial progenitors (bRGs), or postmitotic neurons. Postmitotic neurons migrate along the radial glial fibers or the bRG to the cortical plate [55]. It has been suggested that during cortical expansion, the processes of the aRG are not reaching the whole width of the cortex; therefore, the migrating neurons possibly migrate along non-continuous radial fibers generated from the bRG [56]. The expanded proliferative area of the oSVZ is considered a new niche of progenitors. It is defined by the expression of specific extracellular matrix proteins that can be involved in forming folds in the human brain [55,57]. The considerable expansion of the outer subventricular zone (oSVZ) is of evolutionary significance in the increase in brain volume as well as in the appearance of cortical gyrification [56,58,59,60,61,62,63,64].

Notch is a central signaling pathway that acts at several key time points during corticogenesis, ensuring progenitor pool expansion and maintenance, differentiation and neurogenesis, as well as folding of the cortex. The Notch pathway was first identified in the fruit fly and is highly conserved [65]. Decades of studies have expanded our understanding of the canonical and noncanonical Notch signaling events, its downstream target genes, regulation, and interplay with other signaling pathways (for recent reviews, see [66,67,68]). In a simplified overview, the canonical Notch involves juxtapositioned signal-sending cells that express the bound ligands of the Delta/Serrate/Lag2 family and a receiving cell that expresses the membrane-bound Notch receptors (Notch1-4). The canonical pathway is direct, namely, following ligand binding, the receptor itself undergoes three cleavages, and its intracellular domain (NICD) directly translocates into the nucleus, releasing corepressor complexes (CSL, CBF-1/suppressor of hairless/Lag1), recruits transcription coactivators (MAMLs, Mastermind-like proteins), and promotes target genes expression. The NICD can remain in the cytoplasm, where it will crosstalk with other signaling pathways [69].

Early studies showed that forced activation of Notch signaling (NICD expression) in the mouse forebrain promoted radial glia identity [70], pointing at Notch’s role during the NEC transition to RG. More recently, in human studies, developmental trajectories based on scRNA seq analysis at early neurogenic timepoints recorded the transcriptional transition from early to mature progenitor populations linking it to the disappearance of the Notch1 inhibitor DLK1 [71].

Notch activity suppresses neurogenesis and promotes the maintenance of neural precursors by activating canonical Notch target genes, Hairy/Enhancer of Split (HES), and Hairy/Enhancer of Split related to the YRPW motif (HEY) [68]. This well-establish notion holds for the significant proliferative zone in the human cortex, the osVZ. oRGs express the Notch effector HES1 and undergo neuronal differentiation following pharmacological inhibition of Notch signaling [71]. Interestingly, Notch signaling in progenitors displays an oscillatory behavior that is critical for its ability to sustain them in the proliferative state [72,73]. Being at the crossroad of fundamental developmental events requires the Notch regulatory network to be intrinsicaly robust and coordinated with other processes. We found that Shootin1 plays a dual role as a Notch activator and that the polarity regulator allows radial migrating cells to transit from a multipolar to a bipolar morphology. The enhancing activity of Shootin involves the promotion of the removal of the Notch inhibitor Numb (via interaction with LNX1) and the protection of NICD from degradation (via interaction with Itch) [74].

Finally, Notch activity was linked to the evolutionary expansion and folding of the human cortex. Disturption of neurogenic symmetry, regardless of bRG expansion, is suggested to allow cortical folding. This is achieved by local disruption of Notch activity while maintaining niches of sustain Notch activity [75]. Three paralogs of human-specific NOTCH2NL were found to be highly expressed in RG and were able to sustain their proliferative state, suggesting a possible contribution to human brain evolution [76,77,78,79].

As radial migration proceeds, early-born neurons will generally contribute to the deep layers, whereas late neurons will migrate through the earlier neurons to form the more superficial layers [80]. This organization is known as an “inside-out” organization, wherein newly generated neurons display a multipolar morphology regulated by subplate neurons. These multipolar neurons migrate slowly in the subventricular and intermediate zones [81]. Approximately 80% of the neurons in the cerebral cortex that compose the population of the excitatory or pyramidal neurons follow this radial migration path [55,56,82,83,84]. The inhibitory or GABAergic neurons are born in the GE and travel longer, tangential, migratory routes. Once the GABAergic neurons reach the cortex, they adopt a radial migration pattern. Their position in the cortical layer matches that of the pyramidal neurons born at a particular birthdate [85]. In humans, there is an expanded oSVZ also in the GE, which is particularly marked by the increased size of the caudal ganglionic eminence. These features allow for a higher proportion of interneurons that populate the human brain [84,86]. Another human-specific phenomenon is the continuation of an extensive migration of interneurons during the early postnatal period, which has not been observed in rodents [87].

### 1.4. Human Diseases Associated with Forebrain Development

#### 1.4.1. Holoprosencephaly

One group of human genetic diseases that exhibits aberrant signaling related to brain patterning is holoprosencephaly (HPE) [88,89,90,91,92,93]. HPE is the most common malformation of the forebrain in humans, with an embryonic prevalence of 1 to 250 and a live births prevalence of 1 in 10,000–20,000. HPE is manifested by an incomplete cleavage of the forebrain (prosencephalon) into the right and left hemispheres [93]. The disease is very heterogeneous in appearance and severity, and multiple genes and environmental factors are implicated. This condition can be detected usually at low frequency in many domestic animals such as sheep, horses, and cattle, and in extreme cases can be manifested as cyclopia (single-eye). During the 1950s, there were episodic cases of cycloptic lambs that, in some cases, appeared in up to 25% of a flock’s lambs [94,95]. Due to the economic consequences of these malformations, a lengthy investigative process excluded genetic causes and eventually attributed these cases to the first identified Hedgehog-signaling inhibitor, cyclopamine, found in Veratrum californicum plants in the area [96,97]. Over the years, additional environmental factors that can lead to HPE were identified [98]. These included pesticides such as piperonyl butoxide [99], cannabis-derived phytocannabinoids [100], and exposure to retinoid compounds [101]. Alcohol can also be considered an important contributor to HPE and other congenital disabilities [102].

The genetic etiology of HPE is highly heterogenous and may exhibit a highly variable phenotype. Different cytogenic abnormalities are a leading cause of HPE among live-born patients, and the most prominent ones are trisomy 13 and trisomy 18 [93]. Mutations in members of at least four families of morphogens, SHH, FGF, NODAL, and BMP, are involved in the etiology of HPE [103,104,105] (Figure 3).

The four major mutated genes in HPE are SHH, ZIC2, SIX3, and FGFR1 [93]. SHH was the first HPE gene identified and is the most common one to be mutated [106,107]. Some carriers can be asymptomatic even within the same pedigree, while others exhibit a mild or severe phenotype [93]. It has been hypothesized that mutant proteins can act as modifiers in some cases, and an “autosomal dominant with modifier” model has been suggested [108]. A recent study highlighted two such modifier genes that are positive regulators of SHH signaling. These two genes are highly expressed in LRP2-deficient FVB/N mice, preventing HPE. Both modulators, ULK4 and PTTG1, are microtubule-associated protein components of primary cilia in the neuroepithelium [109]. As another example, among the SHH coreceptors, mutations in CDON and GAS1 were identified to play a role in HPE [110,111,112,113,114,115]; however, mutations in the BOC gene, an SHH gene coreceptor, can modify the observed phenotype [116,117,118,119].

Mutations in the transcription factor ZIC2 are common in HPE. Mice with hypomorphic alleles exhibit middle interhemispheric variant HPE [120,121].

ZIC2 has been suggested to act downstream of NODAL, a secreted member of the transforming growth factor ß (TGF-ß) family, as it is essential for stabilizing the epiblast state and the specification of mesoderm and endoderm, including the PrCP mesoderm, whose instructive functions were discussed. Nodal signals as a heterodimer with TGF-ß family members GDF1/3 and the activated receptor complex phosphorylates the signal transducers Smad2/3 at their C-terminal. ZIC2 possibly acts by direct interaction with SMAD2 and SMAD3 [122]. In addition to its early roles, ZIC2 is working at later stages of brain development and may be involved in limiting Hedgehog signaling [123].

Mutations in SIX3 are another leading cause of HPE [124]. SIX3 is involved in the expression of SHH in the rostral diencephalon ventral midline (in zebrafish) and does so by direct binding to a remote SHH enhancer element (in mice) [125,126]. Mutations in FGFR1 can result in multiple diseases, including HPE [127,128]. FGFR1 belongs to the tyrosine kinase receptor superfamily and contains an extracellular ligand-binding domain and a cytoplasmic domain bearing tyrosine kinase activity. As mentioned above, FGF signaling maintains Shh expression in the prechordal tissue, where it plays a crucial role in the induction of the ventral forebrain [129,130]. Examples of the complex genetics of HPE include documented cases where double mutations in two different genes were reported [131,132]. One of the cases involved a deleterious FGFR1 allele transmitted from one parent and a loss-of-function allele in FGF8 from the other parent [132]. Several such cases with two mutations included the following combinations: FGF8/FGFR1, FGF8/DLL1, DLL1/SHH, DISP1/DISP1, and DISP1/SUFU [128].

Microdeletions and missense mutations in 5′-TG-3′-interacting factor (TGIF) also result in HPE. TGIF interacts with several pathways that are important in the developing brain. TGIF regulates the TGFb/Nodal signaling pathway and SHH signaling independently [133]. In addition, TGIF can function as a repressor to regulate RA-responsive genes [134,135].

BMP signaling has a clear role in brain patterning. Nevertheless, so far, there are no specific mutations in HPE patients related to this pathway. However, several mouse mutants are exhibiting HPE-like phenotypes with mutations in genes associated with this signaling pathway. These mutations include BMP receptors [136], chordin and noggin, antagonists of BMPs in the developing mammalian head [137,138], and mutations in Twisted gastrulation (Tsg) that regulates the pathway through interactions with BMP and chordin [139,140]. In addition, HPE-like phenotypes were observed in mice lacking megalin, which resulted in enhanced Bmp4 expression [141,142].

Mutations in additional genes causing HPE include proteins with several known functions that are likely to regulate the signaling pathways, as mentioned earlier. However, direct connections are yet to be identified. Mutations in CNOT1 that encode for a subunit of the CCR4–NOT Transcription Complex Subunit 1 result in HPE [143,144]. Mutations in members of the cohesin complex of proteins (cohesinopathy), including mutations in STAG2, SMC1A, RAD21, and SMC3, all result in HPE [145,146,147,148]. Loss of function, de novo mutations in protein phosphatase 1 and regulatory subunit 12a (PPP1R12A), an important developmental gene involved in cell migration, adhesion, and morphogenesis, were also associated with HPE [149]. In addition, mutations in the histone lysine methyltransferase 2D (KMT2D) have also been detected in HPE patients [150,151]. In summary, holoprosencephaly can be manifested as an extreme condition. The pathophysiology of the disease process converges on multiple signaling pathways, most of which directly participate in brain patterning.

#### 1.4.2. Retinoic Acid Signaling and Neurodevelopmental Disorders

Consistent with the known functions of RA, loss of RA signaling in the developing fetus has detrimental effects on early arealization and later neurodevelopmental processes. Mutations in genes involved in RA signaling have been implicated in a wide range of diseases, including cancer, metabolic disorders, eye development, retinal function, and neurodegenerative diseases [34,152]. Mutations and gene dosage variation in downstream effectors of RA signaling are implicated in severe developmental syndromes. RAI1 encodes for the transcription factor retinoic acid-induced 1 protein (RAI1). It is involved in both Smith–Magenis syndrome (SMS) and Potocki–Lupski syndrome (PTLS), in which deletions or duplications encompassing the gene were identified, respectively. SMS is a complex disorder characterized by developmental delay, cognitive impairment, and atypical behavioral phenotype [153]. Individuals that suffer from PTLS syndrome have delayed development, mild-to-moderate intellectual disability, behavioral problems, and a high incidence of autism spectrum disorders [154].

Interestingly, abnormal RA signaling is not exclusively the result of a genetic mutation. Exposure of developing fetuses to alcohol is known to cause congenital disabilities and intellectual and neurodevelopmental disabilities. Fetal alcohol syndrome (FASD) was first described in the early 1970s as a specific cluster of congenital disabilities resulting from chronic prenatal alcohol exposure. In the case of the severe form of FASD, the patients have craniofacial malformations (“sentinel facial features”), a high prevalence of microcephaly, prenatal and postnatal growth restriction, and central nervous system neurodevelopmental abnormalities [155]. FASD is considered the leading preventable cause of congenital disabilities and intellectual and neurodevelopmental disabilities, with an estimated global prevalence rate of 0.77%, much higher rates in Europe and North America (1–3%), and lowest rates (0.2%) in countries where religious customs of alcohol abstinence are common. Despite this alarming data, the numbers are considered an underestimation, as the drinking habits of pregnant mothers are often not disclosed [155]. Evidence linking reduced RA signaling during embryogenesis to in utero exposure to alcohol has provided one of the prominent etiological models for the syndrome. Other suggested disease mechanisms consider the maleffects of ethanol on cellular and subcellular levels. As discussed earlier, ethanol itself, rather than its metabolites, is regarded as an HPE-inducing teratogen and can act as a modifier that synergizes with loss-of-function mutations in the SHH signaling pathway [116,156]. An alternative, or rather a complementary model, namely, the reduction of RA during early development, can be linked to multiple disease phenotypes and explain the severe teratogenic outcome of prenatal alcohol exposure [157,158].

The connection between ethanol metabolism and retinol arises from the biochemical similarity between ethanol clearance from the body and RA biosynthesis [158,159]. Ethanol competes for the enzymes with alcohol dehydrogenase activity, suggesting that ethanol or its oxidation metabolite, retinaldehyde, can competitively inhibit the production of retinoic acid [159]. In fact, acetaldehyde is the preferred substrate of RALDH2, one of two retinaldehyde dehydrogenases that oxidize retinaldehyde to form retinoic acid.

#### 1.4.3. Lissencephaly with Regional Gradients of Severity

In the mature brain, the areal organization can be manifested in differences in the cytoarchitecture, cell number, cell density, and lamination in different brain areas. Histological differences define anatomical subdivisions rooted in the developmental phases of forebrain development. They may result in differential sensitivity to pathological mutations or variations in gene dosage that dictate the disease phenotype.

The lissencephaly–pachygyria spectrum of diseases defines a variety of brain malformations that cause relative smoothness of the brain surface and includes lissencephaly (smooth brain surface), agyria (no gyri), and pachygyria (broad gyri) (Figure 4). These brain malformations are partially due to the impairment of neuron migration in the developing brain. Only four layers can be observed instead of the normal six layers in the cortex [160,161].

Several of the lissencephalies exhibit area-specific severity of the phenotype [4,162]. In a few cases, the preferential brain region exhibiting a more pronounced phenotype has been attributed to the expression pattern of the mutated gene or its protein interactor [163]. However, in most cases, the regional preference is not understood and requires additional research.

Regional severity has been noted in lissencephaly caused by mutations in *LIS1, Lissencephaly 1* [164], or *DCX, Doublecortin* [165,166]. *LIS1* mutations result in a more severe phenotype in the parietal and occipital cortex, whereas in the case of mutations in *DCX*, the malformation is more pronounced in the frontal cortex [167,168]. A smooth cerebral surface with a posterior-to-anterior gradient of severity was noted in cases of mutations in *ARX* [169,170], *KIF2A* [171], *MACF1* [172], *DYNC1H1* [173,174,175]. The same trend was noted for mutations in several tubulin genes; *TUBA1A* [176,177,178], *TUBG1* [179], *TUBB2B, TUBB3* [180], *TUBB5* [181], *TUBA8* [182]. Mutations in the centrosomal-associated protein CEP85L result in posterior predominant lissencephaly [183]. A different study demonstrated that the severity gradient reflected the expression pattern and that CEP85L is required to localize and activate the phosphorylation of CDK5 at the centrosome [184].

Anterior-to-posterior increased severity also involves genes of the actin superfamily of proteins: *ACTG1* [185], *ACTB* [185,186], *CRADD* [187], and *RTTN* [188].

## 2. Concluding Remarks

Understanding and identifying developmental principles and signaling pathways that shape the anterior CNS have progressed considerably. The areal organization of the human brain is tightly linked to brain function. Thus, the proper progression of the developmental plan is critical for brain functions. A deeper understanding of the etiology of neurodevelopmental conditions is rooted in uncovering the essential steps during brain patterning. This may be true not only when gross brain abnormalities are seen but also when abnormalities are manifested in subtle arealization defects that are not always appreciated but can profoundly influence an individual’s cognitive and social skills.

## Figures and Tables

**Figure 1 cells-11-01642-f001:**
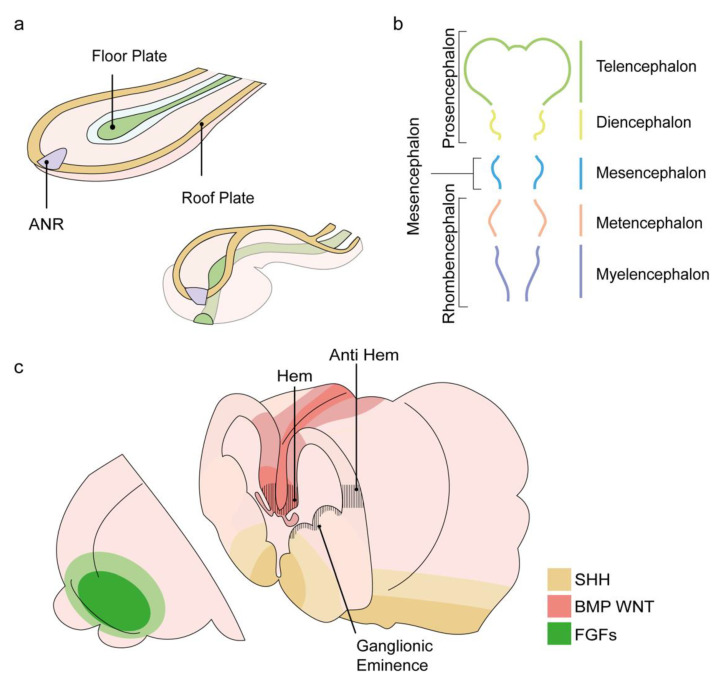
Schematic presentation of the events leading to forebrain formation. (**a**) Rostro-lateral representation of the neural plate as it folds and fuses at the midline to form the neural tube. The anterior neural ridge (ANR), shown in purple, is an organizer of the forebrain. Positioned at the most rostral part of the neural plate, its signaling will promote the formation of the future forebrain structure, the prosencephalon. Modified from [6]. (**b**) The primary vesicles appear in the fourth week of development and introduce three morphological segments of the brain: the prosencephalon, the mesencephalon, and the rhombencephalon. By the following week, the prosencephalon will further subdivide into the future forebrain structures: the anterior neural tube, the two telencephalon vesicles, and the diencephalon. (**c**) Sources of key morphogens are indicated; FGFs in green, BMPs and WNTs in red, SHH in yellow. The positions of the hem, anti-hem, and ganglionic eminences are indicated with black lines. Modified from [7].

**Figure 2 cells-11-01642-f002:**
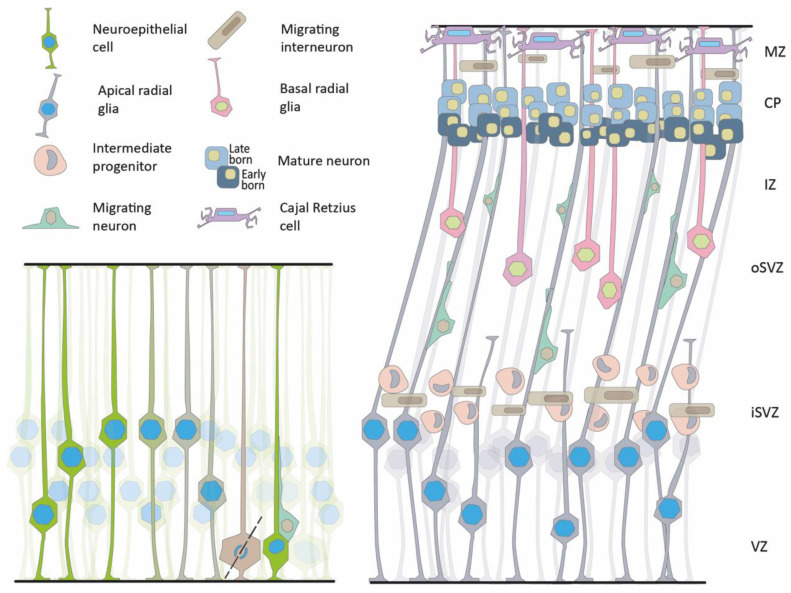
The neuroepithelium and neuronal migration. Left, schematic presentation of the early neuroepithelium, showing interkinetic nuclear motility and asymmetric cell division leading to the formation of two daughter cells, a progenitor and a migrating neuron. Right, Later stages of brain development show three proliferative areas, the ventricular zone (VZ), the inner subventricular zone (iSVZ), and the outer subventricular zone (oSVZ). Different types of progenitors can be detected in the proliferative zones, radial glia, intermediate progenitors, and basal radial glia. Migrating neurons can be seen in the intermediate zone (IZ). The pyramidal neurons are organized in layers in the forming cortical plate (CP) based on their birth date. Cajal–Retzius cells that are born the earliest are detected in the marginal zone (MZ). Migrating interneurons are visible in two migratory streams.

**Figure 3 cells-11-01642-f003:**
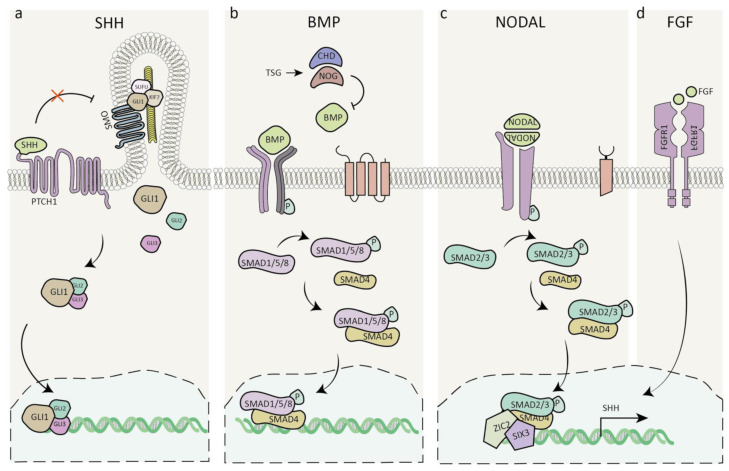
Signalling pathways involved in holoprosencephaly. (**a**) SHH signaling pathway. (**b**) BMP antagonists TSG/CHD/NOG suppress the activation of BMP downstream effectors. (**c**) Nodal signaling leads to the phosphorylation of smad2/3, which translocates to the nucleus, where it is thought to interact with ZIC2 and regulate SHH expression. (**d**) FGF signaling maintains SHH expression. Abbreviations: SHH, Sonic hedgehog; GLI1-3, glioma-associated oncogene homolog; PTCH1, Patched1; SMO, Smoothened; SUFU, Suppressor of fused homolog; KIF7, Kinesin family member 7; TSG, Twisted gastrulation; CHD, Chordin; NOG, Noggin; BMP, Bone morphogenic protein; P, Phosphate; SMAD, Mothers against decapentaplegic homolog; NODAL, Nodal growth differentiation factor; ZIC2, Zinc finger protein 2; SIX3, SIX homeobox 3; FGF/FGFR1, Fibroblast growth factor/receptor 1.

**Figure 4 cells-11-01642-f004:**
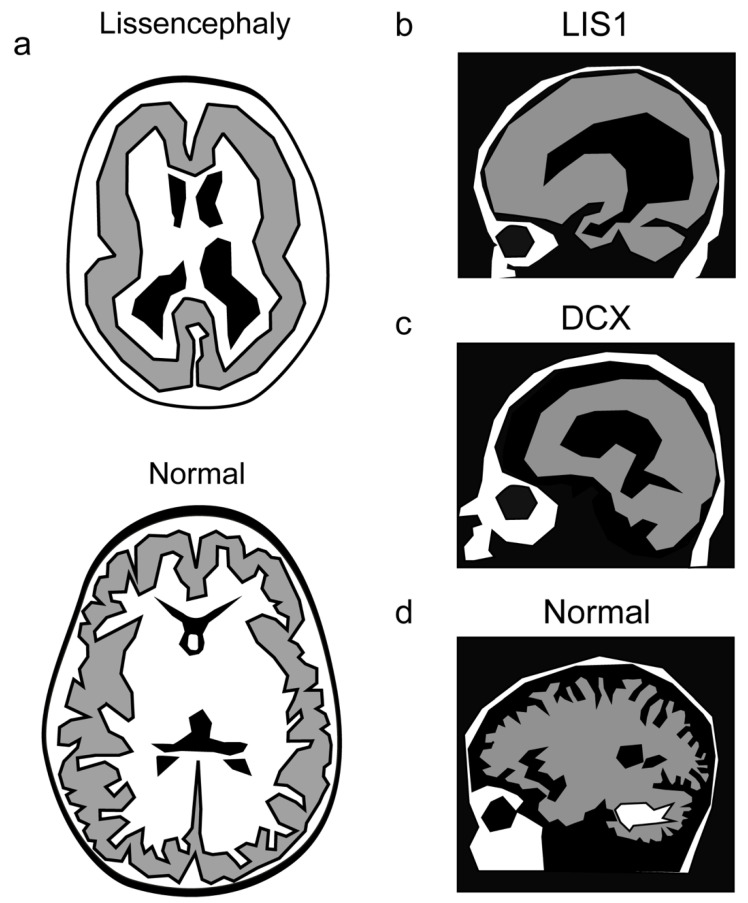
Lissencephaly manifests with a reduction in the normal brain folds. (**a**) A schematic of a lissencephalic and a normal brain. (**b**–**d**) Mutations in LIS1 result in a more severe phenotype in the caudal part of the brain (**b**), whereas mutations in DCX affect more the rostral part of the brain (**c**), as shown in comparison with the normal brain (**d**).

## Data Availability

Not applicable.

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
