# Peer review of "Brain Organization and Human Diseases"

_cells, 2022, doi:10.3390/cells11101642_

Round 1

Reviewer 1 Report

The manuscript by Sapir et al., is a review describing forebrain patterning during human embryonic development, the role of signaling molecules controlling forebrain development, and migration of neurons and formation of cortical layers. The second part of the review focused on diseases causing defective brain patterning, the role of retinoic acid signaling, and neurodevelopmental disorders and lissencephalopathy. The review is very interesting for readers of Cells and summarizes aspects of brain development and patterning as well as malformations after mutation of signaling molecules or disorders of signaling.

The manuscript was comprehensible written and I did not find obvious errors or omissions in the manuscript.

Minor point:

Please can you check again to spell in full abbreviations at their first use (please confer line 76 and line 150).

Author Response

Please can you check again to spell in full abbreviations at their first use (please confer line 76 and line 150).

We thank the reviewer for noting this!

In addition to the above, we added a section related to BMP signaling and corrected a few typos.

BMP signaling has a clear role in brain patterning. Nevertheless, so far, there are no specific mutations in HPE patients related to this pathway. However, several mouse mutants are exhibiting HPE-like phenotypes with mutations in genes associated with this signaling pathway. These mutations include BMP receptors [15], chordin and noggin, antagonists of BMPs in the developing mammalian head [16,17], and mutations in Twisted gastrulation (Tsg) that regulates the pathway through interactions with BMP and chordin [18,19]. In addition, HPE-like phenotypes were observed in mice lacking megalin, which resulted in enhanced Bmp4 expression [20,21].

Additional references

  1. Klingler, E.; Francis, F.; Jabaudon, D.; Cappello, S. Mapping the molecular and cellular complexity of cortical malformations. Science 2021, 371, doi:10.1126/science.aba4517.
  2. Francis, F.; Cappello, S. Neuronal migration and disorders - an update. Curr Opin Neurobiol 2021, 66, 57-68, doi:10.1016/j.conb.2020.10.002.
  3. Borggrefe, T.; Lauth, M.; Zwijsen, A.; Huylebroeck, D.; Oswald, F.; Giaimo, B.D. The Notch intracellular domain integrates signals from Wnt, Hedgehog, TGFbeta/BMP and hypoxia pathways. Biochim Biophys Acta 2016, 1863, 303-313, doi:10.1016/j.bbamcr.2015.11.020.
  4. Gaiano, N.; Fishell, G. The role of notch in promoting glial and neural stem cell fates. Annu Rev Neurosci 2002, 25, 471-490, doi:10.1146/annurev.neuro.25.030702.130823.
  5. Eze, U.C.; Bhaduri, A.; Haeussler, M.; Nowakowski, T.J.; Kriegstein, A.R. Single-cell atlas of early human brain development highlights heterogeneity of human neuroepithelial cells and early radial glia. Nat Neurosci 2021, 24, 584-594, doi:10.1038/s41593-020-00794-1.
  6. Zhou, B.; Lin, W.; Long, Y.; Yang, Y.; Zhang, H.; Wu, K.; Chu, Q. Notch signaling pathway: architecture, disease, and therapeutics. Signal Transduct Target Ther 2022, 7, 95, doi:10.1038/s41392-022-00934-y.
  7. Kageyama, R.; Shimojo, H.; Isomura, A. Oscillatory Control of Notch Signaling in Development. Adv Exp Med Biol 2018, 1066, 265-277, doi:10.1007/978-3-319-89512-3_13.
  8. Kageyama, R.; Shimojo, H.; Ohtsuka, T. Dynamic control of neural stem cells by bHLH factors. Neurosci Res 2019, 138, 12-18, doi:10.1016/j.neures.2018.09.005.
  9. Sapir, T.; Levy, T.; Kozer, N.; Shin, I.; Zamor, V.; Haffner-Krausz, R.; McGlade, J.C.; Reiner, O. Notch Activation by Shootin1 Opposing Activities on 2 Ubiquitin Ligases. Cereb Cortex 2018, 28, 3115-3128, doi:10.1093/cercor/bhx180.
  10. Han, S.; Okawa, S.; Wilkinson, G.A.; Ghazale, H.; Adnani, L.; Dixit, R.; Tavares, L.; Faisal, I.; Brooks, M.J.; Cortay, V.; et al. Proneural genes define ground-state rules to regulate neurogenic patterning and cortical folding. Neuron 2021, doi:10.1016/j.neuron.2021.07.007.
  11. Lodewijk, G.A.; Fernandes, D.P.; Vretzakis, I.; Savage, J.E.; Jacobs, F.M.J. Evolution of Human Brain Size-Associated NOTCH2NL Genes Proceeds toward Reduced Protein Levels. Mol Biol Evol 2020, 37, 2531-2548, doi:10.1093/molbev/msaa104.
  12. Fiddes, I.T.; Lodewijk, G.A.; Mooring, M.; Bosworth, C.M.; Ewing, A.D.; Mantalas, G.L.; Novak, A.M.; van den Bout, A.; Bishara, A.; Rosenkrantz, J.L.; et al. Human-Specific NOTCH2NL Genes Affect Notch Signaling and Cortical Neurogenesis. Cell 2018, 173, 1356-1369 e1322, doi:10.1016/j.cell.2018.03.051.
  13. Florio, M.; Heide, M.; Pinson, A.; Brandl, H.; Albert, M.; Winkler, S.; Wimberger, P.; Huttner, W.B.; Hiller, M. Evolution and cell-type specificity of human-specific genes preferentially expressed in progenitors of fetal neocortex. Elife 2018, 7, doi:10.7554/eLife.32332.
  14. Suzuki, I.K.; Gacquer, D.; Van Heurck, R.; Kumar, D.; Wojno, M.; Bilheu, A.; Herpoel, A.; Lambert, N.; Cheron, J.; Polleux, F.; et al. Human-Specific NOTCH2NL Genes Expand Cortical Neurogenesis through Delta/Notch Regulation. Cell 2018, 173, 1370-1384 e1316, doi:10.1016/j.cell.2018.03.067.
  15. Fernandes, M.; Gutin, G.; Alcorn, H.; McConnell, S.K.; Hebert, J.M. Mutations in the BMP pathway in mice support the existence of two molecular classes of holoprosencephaly. Development 2007, 134, 3789-3794, doi:10.1242/dev.004325.
  16. Anderson, R.M.; Lawrence, A.R.; Stottmann, R.W.; Bachiller, D.; Klingensmith, J. Chordin and noggin promote organizing centers of forebrain development in the mouse. Development 2002, 129, 4975-4987, doi:10.1242/dev.129.21.4975.
  17. Bachiller, D.; Klingensmith, J.; Kemp, C.; Belo, J.A.; Anderson, R.M.; May, S.R.; McMahon, J.A.; McMahon, A.P.; Harland, R.M.; Rossant, J.; et al. The organizer factors Chordin and Noggin are required for mouse forebrain development. Nature 2000, 403, 658-661, doi:10.1038/35001072.
  18. Zakin, L.; De Robertis, E.M. Inactivation of mouse Twisted gastrulation reveals its role in promoting Bmp4 activity during forebrain development. Development 2004, 131, 413-424, doi:10.1242/dev.00946.
  19. Petryk, A.; Anderson, R.M.; Jarcho, M.P.; Leaf, I.; Carlson, C.S.; Klingensmith, J.; Shawlot, W.; O'Connor, M.B. The mammalian twisted gastrulation gene functions in foregut and craniofacial development. Dev Biol 2004, 267, 374-386, doi:10.1016/j.ydbio.2003.11.015.
  20. Willnow, T.E.; Hilpert, J.; Armstrong, S.A.; Rohlmann, A.; Hammer, R.E.; Burns, D.K.; Herz, J. Defective forebrain development in mice lacking gp330/megalin. Proc Natl Acad Sci U S A 1996, 93, 8460-8464, doi:10.1073/pnas.93.16.8460.
  21. Spoelgen, R.; Hammes, A.; Anzenberger, U.; Zechner, D.; Andersen, O.M.; Jerchow, B.; Willnow, T.E. LRP2/megalin is required for patterning of the ventral telencephalon. Development 2005, 132, 405-414, doi:10.1242/dev.01580.

Reviewer 2 Report

Sapir et al. review on the brain organization and human diseases. This is a comprehensive review describing how brain is developed, and what human disorders develop if normal process of brain development is disturbed. Holoprocencephaly is a nice example because not only genetic but environmental factors could cause the disorder, highlighting the interaction of both factors on the same signaling pathway.

I have only a few comments.

  • Since many genes are involved in the development of human brain malformation including holoprocencephaly, a schematic figure showing the pathway and target sites of gene products would help understand the interaction of each gene.
  • I found some typos in page 4, line 141-144.

Author Response

  • Since many genes are involved in the development of human brain malformation including holoprosencephaly, a schematic figure showing the pathway and target sites of gene products would help understand the interaction of each gene.

         We added a new figure illustrating this point.

  • I found some typos in page 4, line 141-144.

         corrected.

In addition to the above, we added a section related to BMP signaling and corrected a few typos.

BMP signaling has a clear role in brain patterning. Nevertheless, so far, there are no specific mutations in HPE patients related to this pathway. However, several mouse mutants are exhibiting HPE-like phenotypes with mutations in genes associated with this signaling pathway. These mutations include BMP receptors [15], chordin and noggin, antagonists of BMPs in the developing mammalian head [16,17], and mutations in Twisted gastrulation (Tsg) that regulates the pathway through interactions with BMP and chordin [18,19]. In addition, HPE-like phenotypes were observed in mice lacking megalin, which resulted in enhanced Bmp4 expression [20,21].

Additional references

  1. Klingler, E.; Francis, F.; Jabaudon, D.; Cappello, S. Mapping the molecular and cellular complexity of cortical malformations. Science 2021, 371, doi:10.1126/science.aba4517.
  2. Francis, F.; Cappello, S. Neuronal migration and disorders - an update. Curr Opin Neurobiol 2021, 66, 57-68, doi:10.1016/j.conb.2020.10.002.
  3. Borggrefe, T.; Lauth, M.; Zwijsen, A.; Huylebroeck, D.; Oswald, F.; Giaimo, B.D. The Notch intracellular domain integrates signals from Wnt, Hedgehog, TGFbeta/BMP and hypoxia pathways. Biochim Biophys Acta 2016, 1863, 303-313, doi:10.1016/j.bbamcr.2015.11.020.
  4. Gaiano, N.; Fishell, G. The role of notch in promoting glial and neural stem cell fates. Annu Rev Neurosci 2002, 25, 471-490, doi:10.1146/annurev.neuro.25.030702.130823.
  5. Eze, U.C.; Bhaduri, A.; Haeussler, M.; Nowakowski, T.J.; Kriegstein, A.R. Single-cell atlas of early human brain development highlights heterogeneity of human neuroepithelial cells and early radial glia. Nat Neurosci 2021, 24, 584-594, doi:10.1038/s41593-020-00794-1.
  6. Zhou, B.; Lin, W.; Long, Y.; Yang, Y.; Zhang, H.; Wu, K.; Chu, Q. Notch signaling pathway: architecture, disease, and therapeutics. Signal Transduct Target Ther 2022, 7, 95, doi:10.1038/s41392-022-00934-y.
  7. Kageyama, R.; Shimojo, H.; Isomura, A. Oscillatory Control of Notch Signaling in Development. Adv Exp Med Biol 2018, 1066, 265-277, doi:10.1007/978-3-319-89512-3_13.
  8. Kageyama, R.; Shimojo, H.; Ohtsuka, T. Dynamic control of neural stem cells by bHLH factors. Neurosci Res 2019, 138, 12-18, doi:10.1016/j.neures.2018.09.005.
  9. Sapir, T.; Levy, T.; Kozer, N.; Shin, I.; Zamor, V.; Haffner-Krausz, R.; McGlade, J.C.; Reiner, O. Notch Activation by Shootin1 Opposing Activities on 2 Ubiquitin Ligases. Cereb Cortex 2018, 28, 3115-3128, doi:10.1093/cercor/bhx180.
  10. Han, S.; Okawa, S.; Wilkinson, G.A.; Ghazale, H.; Adnani, L.; Dixit, R.; Tavares, L.; Faisal, I.; Brooks, M.J.; Cortay, V.; et al. Proneural genes define ground-state rules to regulate neurogenic patterning and cortical folding. Neuron 2021, doi:10.1016/j.neuron.2021.07.007.
  11. Lodewijk, G.A.; Fernandes, D.P.; Vretzakis, I.; Savage, J.E.; Jacobs, F.M.J. Evolution of Human Brain Size-Associated NOTCH2NL Genes Proceeds toward Reduced Protein Levels. Mol Biol Evol 2020, 37, 2531-2548, doi:10.1093/molbev/msaa104.
  12. Fiddes, I.T.; Lodewijk, G.A.; Mooring, M.; Bosworth, C.M.; Ewing, A.D.; Mantalas, G.L.; Novak, A.M.; van den Bout, A.; Bishara, A.; Rosenkrantz, J.L.; et al. Human-Specific NOTCH2NL Genes Affect Notch Signaling and Cortical Neurogenesis. Cell 2018, 173, 1356-1369 e1322, doi:10.1016/j.cell.2018.03.051.
  13. Florio, M.; Heide, M.; Pinson, A.; Brandl, H.; Albert, M.; Winkler, S.; Wimberger, P.; Huttner, W.B.; Hiller, M. Evolution and cell-type specificity of human-specific genes preferentially expressed in progenitors of fetal neocortex. Elife 2018, 7, doi:10.7554/eLife.32332.
  14. Suzuki, I.K.; Gacquer, D.; Van Heurck, R.; Kumar, D.; Wojno, M.; Bilheu, A.; Herpoel, A.; Lambert, N.; Cheron, J.; Polleux, F.; et al. Human-Specific NOTCH2NL Genes Expand Cortical Neurogenesis through Delta/Notch Regulation. Cell 2018, 173, 1370-1384 e1316, doi:10.1016/j.cell.2018.03.067.
  15. Fernandes, M.; Gutin, G.; Alcorn, H.; McConnell, S.K.; Hebert, J.M. Mutations in the BMP pathway in mice support the existence of two molecular classes of holoprosencephaly. Development 2007, 134, 3789-3794, doi:10.1242/dev.004325.
  16. Anderson, R.M.; Lawrence, A.R.; Stottmann, R.W.; Bachiller, D.; Klingensmith, J. Chordin and noggin promote organizing centers of forebrain development in the mouse. Development 2002, 129, 4975-4987, doi:10.1242/dev.129.21.4975.
  17. Bachiller, D.; Klingensmith, J.; Kemp, C.; Belo, J.A.; Anderson, R.M.; May, S.R.; McMahon, J.A.; McMahon, A.P.; Harland, R.M.; Rossant, J.; et al. The organizer factors Chordin and Noggin are required for mouse forebrain development. Nature 2000, 403, 658-661, doi:10.1038/35001072.
  18. Zakin, L.; De Robertis, E.M. Inactivation of mouse Twisted gastrulation reveals its role in promoting Bmp4 activity during forebrain development. Development 2004, 131, 413-424, doi:10.1242/dev.00946.
  19. Petryk, A.; Anderson, R.M.; Jarcho, M.P.; Leaf, I.; Carlson, C.S.; Klingensmith, J.; Shawlot, W.; O'Connor, M.B. The mammalian twisted gastrulation gene functions in foregut and craniofacial development. Dev Biol 2004, 267, 374-386, doi:10.1016/j.ydbio.2003.11.015.
  20. Willnow, T.E.; Hilpert, J.; Armstrong, S.A.; Rohlmann, A.; Hammer, R.E.; Burns, D.K.; Herz, J. Defective forebrain development in mice lacking gp330/megalin. Proc Natl Acad Sci U S A 1996, 93, 8460-8464, doi:10.1073/pnas.93.16.8460.
  21. Spoelgen, R.; Hammes, A.; Anzenberger, U.; Zechner, D.; Andersen, O.M.; Jerchow, B.; Willnow, T.E. LRP2/megalin is required for patterning of the ventral telencephalon. Development 2005, 132, 405-414, doi:10.1242/dev.01580.

Reviewer 3 Report

In the article "Brain organization and human diseases" Sapir and colleagues deal with a topic of high scientific interest: how brain formation occurs during development and what may be the causes of cortical malformations that generate human diseases. The topic is well covered and the article well structured. Nonetheless, I have some criticisms for the authors:

1- another Review has recently been published that deals with the same topic (Mapping the molecular and cellular complexity of
cortical malformations, from Klingler et al., 2021). In addition to the fact that this article is not even mentioned in this article, what does your article bring about innovative compared to that of Klingler and colleagues?

2- In paragraph 2.2. "Signaling factors that control forebrain development", there is no mention of the Notch pathway, which plays a key role in the process of brain formation (see also the review by Nian and Hou, 2022). Notch pathway is only briefly mentioned in the next paragraph (2.3), but in my opinion it deserves a dedicated paragraph in chapter 2.2

3- In the text there are some small errors that must be corrected: 1- fig. 1, replace figure ledend with figure legend; 2- line 138, replace AR with RA; 3- line 141, replace thatsignaling with that signaling; 4- line 144, replace downstream transcription with downstream transcription.

Author Response

1.  another Review has recently been published that deals with the same topic (Mapping the molecular and cellular complexity of
cortical malformations, from Klingler et al., 2021). In addition to the fact that this article is not even mentioned in this article, what does your article bring about innovative compared to that of Klingler and colleagues?

We thank the reviewer for pointing out that we lacked to quote this review. We indeed added this review and an additional one covering some of the aspects mentioned in our review.

Nevertheless, our review covers additional topics and brain malformations not mentioned in these two excellent reviews. Our review tried to provide a comprehensive outline of how the forebrain develops from the early neural plate. To highlight some of the initial morphogens and signaling pathways involved and indicate some brain malformations related to these pathways, such as holoprosencephaly. Our review does not cover abnormal networks and behavioral defects discussed in the review by Klingler et al.[1].

[1,2]

2.  In paragraph 2.2. "Signaling factors that control forebrain development", there is no mention of the Notch pathway, which plays a key role in the process of brain formation (see also the review by Nian and Hou, 2022). Notch pathway is only briefly mentioned in the next paragraph (2.3), but in my opinion it deserves a dedicated paragraph in chapter 2.2

We added a new section regarding the Notch pathway.

Notch involves juxtapositioned signal sending cells that express the bound ligands of the Delta/Serrate/Lag2 family and a receiving cell that expresses the membrane-bound Notch receptors (Notch1-4). The canonical pathway is direct, namely, following ligand binding, the receptor itself undergoes three cleavages, and its intracellular domain (NICD) directly translocates into the nucleus, releasing corepressor complexes ( CSL, CBF-1/suppressor of hairless/Lag1), recruits transcription coactivators (MAMLs, Mastermind-like proteins) and promotes target genes expression. The NICD can remain in the cytoplasm, where it will crosstalk with other signaling pathways [3].

Early studies showed that forced activation of Notch signaling (NICD expression) in the mouse forebrain promoted radial glia identity[4], pointing at Notch’s role during the NEC transition to RG. More recently, in human studies, developmental trajectories based on scRNA seq analysis at early neurogenic timepoints record the transcriptional transition from early to mature progenitor populations linking it to the disappearance of a Notch1 inhibitor DLK1 [5].

Notch activity suppresses neurogenesis and promotes the maintenance of neural precursors by activating canonical Notch target genes, Hairy/Enhancer of Split (HES) and Hairy/Enhancer of Split related to YRPW motif (HEY) [6]. This well-establish notion holds for the significant proliferative zone in the human cortex, the osVZ. oRGs express Notch effector, HES1, and undergo neuronal differentiation following pharmacological inhibition of Notch signaling [5]. Interestingly, Notch signaling in progenitors displays an oscillatory behavior that is critical for its ability to sustain them in the proliferative state [7,8]. Being at the crossroad of fundamental developmental events requires Notch regulatory network to be intricate robust, and coordinated with other processes. We found that Shootin1 plays a dual role as a Notch activator and that the polarity regulator allows radial migrating cells to transit from multipolar to bipolar morphology. The enhancing activity of Shootin involves the promotion of the removal of Notch inhibitor, Numb (via interaction with LNX1), and the protection of NICD from degradation (via interaction with Itch)[9].

Finally, Notch activity was linked to the evolutionary expansion and folding of the human cortex. Disturption of neurogenic symmetry, regardless of bRG expansion, is suggested to allow cortical folding. This is achieved by local disruption of Notch activity while maintaining niches of sustain Notch activity [10]. Three paralogs of human-specific NOTCH2NL were found to be highly expressed in RG and were able to sustain their proliferative state, suggesting a possible contribution to human brain evolution [11-14].

3.  In the text there are some small errors that must be corrected: 1- fig. 1, replace figure ledend with figure legend; 2- line 138, replace AR with RA; 3- line 141, replace thatsignaling with that signaling; 4- line 144, replace downstream transcription with downstream transcription.

1- fig. 1, replace figure ledend with figure legend

Removed from the figure.

2- line 138, replace AR with RA;

Corrected.

3- line 141, replace thatsignaling with that signaling

Corrected.

4- line 144, replace downstream transcription with downstream transcription.

Corrected.

In addition to the above, we added a section related to BMP signaling and corrected a few typos.

BMP signaling has a clear role in brain patterning. Nevertheless, so far, there are no specific mutations in HPE patients related to this pathway. However, several mouse mutants are exhibiting HPE-like phenotypes with mutations in genes associated with this signaling pathway. These mutations include BMP receptors [15], chordin and noggin, antagonists of BMPs in the developing mammalian head [16,17], and mutations in Twisted gastrulation (Tsg) that regulates the pathway through interactions with BMP and chordin [18,19]. In addition, HPE-like phenotypes were observed in mice lacking megalin, which resulted in enhanced Bmp4 expression [20,21].

Additional references

  1. Klingler, E.; Francis, F.; Jabaudon, D.; Cappello, S. Mapping the molecular and cellular complexity of cortical malformations. Science 2021, 371, doi:10.1126/science.aba4517.
  2. Francis, F.; Cappello, S. Neuronal migration and disorders - an update. Curr Opin Neurobiol 2021, 66, 57-68, doi:10.1016/j.conb.2020.10.002.
  3. Borggrefe, T.; Lauth, M.; Zwijsen, A.; Huylebroeck, D.; Oswald, F.; Giaimo, B.D. The Notch intracellular domain integrates signals from Wnt, Hedgehog, TGFbeta/BMP and hypoxia pathways. Biochim Biophys Acta 2016, 1863, 303-313, doi:10.1016/j.bbamcr.2015.11.020.
  4. Gaiano, N.; Fishell, G. The role of notch in promoting glial and neural stem cell fates. Annu Rev Neurosci 2002, 25, 471-490, doi:10.1146/annurev.neuro.25.030702.130823.
  5. Eze, U.C.; Bhaduri, A.; Haeussler, M.; Nowakowski, T.J.; Kriegstein, A.R. Single-cell atlas of early human brain development highlights heterogeneity of human neuroepithelial cells and early radial glia. Nat Neurosci 2021, 24, 584-594, doi:10.1038/s41593-020-00794-1.
  6. Zhou, B.; Lin, W.; Long, Y.; Yang, Y.; Zhang, H.; Wu, K.; Chu, Q. Notch signaling pathway: architecture, disease, and therapeutics. Signal Transduct Target Ther 2022, 7, 95, doi:10.1038/s41392-022-00934-y.
  7. Kageyama, R.; Shimojo, H.; Isomura, A. Oscillatory Control of Notch Signaling in Development. Adv Exp Med Biol 2018, 1066, 265-277, doi:10.1007/978-3-319-89512-3_13.
  8. Kageyama, R.; Shimojo, H.; Ohtsuka, T. Dynamic control of neural stem cells by bHLH factors. Neurosci Res 2019, 138, 12-18, doi:10.1016/j.neures.2018.09.005.
  9. Sapir, T.; Levy, T.; Kozer, N.; Shin, I.; Zamor, V.; Haffner-Krausz, R.; McGlade, J.C.; Reiner, O. Notch Activation by Shootin1 Opposing Activities on 2 Ubiquitin Ligases. Cereb Cortex 2018, 28, 3115-3128, doi:10.1093/cercor/bhx180.
  10. Han, S.; Okawa, S.; Wilkinson, G.A.; Ghazale, H.; Adnani, L.; Dixit, R.; Tavares, L.; Faisal, I.; Brooks, M.J.; Cortay, V.; et al. Proneural genes define ground-state rules to regulate neurogenic patterning and cortical folding. Neuron 2021, doi:10.1016/j.neuron.2021.07.007.
  11. Lodewijk, G.A.; Fernandes, D.P.; Vretzakis, I.; Savage, J.E.; Jacobs, F.M.J. Evolution of Human Brain Size-Associated NOTCH2NL Genes Proceeds toward Reduced Protein Levels. Mol Biol Evol 2020, 37, 2531-2548, doi:10.1093/molbev/msaa104.
  12. Fiddes, I.T.; Lodewijk, G.A.; Mooring, M.; Bosworth, C.M.; Ewing, A.D.; Mantalas, G.L.; Novak, A.M.; van den Bout, A.; Bishara, A.; Rosenkrantz, J.L.; et al. Human-Specific NOTCH2NL Genes Affect Notch Signaling and Cortical Neurogenesis. Cell 2018, 173, 1356-1369 e1322, doi:10.1016/j.cell.2018.03.051.
  13. Florio, M.; Heide, M.; Pinson, A.; Brandl, H.; Albert, M.; Winkler, S.; Wimberger, P.; Huttner, W.B.; Hiller, M. Evolution and cell-type specificity of human-specific genes preferentially expressed in progenitors of fetal neocortex. Elife 2018, 7, doi:10.7554/eLife.32332.
  14. Suzuki, I.K.; Gacquer, D.; Van Heurck, R.; Kumar, D.; Wojno, M.; Bilheu, A.; Herpoel, A.; Lambert, N.; Cheron, J.; Polleux, F.; et al. Human-Specific NOTCH2NL Genes Expand Cortical Neurogenesis through Delta/Notch Regulation. Cell 2018, 173, 1370-1384 e1316, doi:10.1016/j.cell.2018.03.067.
  15. Fernandes, M.; Gutin, G.; Alcorn, H.; McConnell, S.K.; Hebert, J.M. Mutations in the BMP pathway in mice support the existence of two molecular classes of holoprosencephaly. Development 2007, 134, 3789-3794, doi:10.1242/dev.004325.
  16. Anderson, R.M.; Lawrence, A.R.; Stottmann, R.W.; Bachiller, D.; Klingensmith, J. Chordin and noggin promote organizing centers of forebrain development in the mouse. Development 2002, 129, 4975-4987, doi:10.1242/dev.129.21.4975.
  17. Bachiller, D.; Klingensmith, J.; Kemp, C.; Belo, J.A.; Anderson, R.M.; May, S.R.; McMahon, J.A.; McMahon, A.P.; Harland, R.M.; Rossant, J.; et al. The organizer factors Chordin and Noggin are required for mouse forebrain development. Nature 2000, 403, 658-661, doi:10.1038/35001072.
  18. Zakin, L.; De Robertis, E.M. Inactivation of mouse Twisted gastrulation reveals its role in promoting Bmp4 activity during forebrain development. Development 2004, 131, 413-424, doi:10.1242/dev.00946.
  19. Petryk, A.; Anderson, R.M.; Jarcho, M.P.; Leaf, I.; Carlson, C.S.; Klingensmith, J.; Shawlot, W.; O'Connor, M.B. The mammalian twisted gastrulation gene functions in foregut and craniofacial development. Dev Biol 2004, 267, 374-386, doi:10.1016/j.ydbio.2003.11.015.
  20. Willnow, T.E.; Hilpert, J.; Armstrong, S.A.; Rohlmann, A.; Hammer, R.E.; Burns, D.K.; Herz, J. Defective forebrain development in mice lacking gp330/megalin. Proc Natl Acad Sci U S A 1996, 93, 8460-8464, doi:10.1073/pnas.93.16.8460.
  21. Spoelgen, R.; Hammes, A.; Anzenberger, U.; Zechner, D.; Andersen, O.M.; Jerchow, B.; Willnow, T.E. LRP2/megalin is required for patterning of the ventral telencephalon. Development 2005, 132, 405-414, doi:10.1242/dev.01580.